# Aptamer-Functionalized Natural Protein-Based Polymers as Innovative Biomaterials

**DOI:** 10.3390/pharmaceutics12111115

**Published:** 2020-11-19

**Authors:** Alessandra Girotti, Sara Escalera-Anzola, Irene Alonso-Sampedro, Juan González-Valdivieso, Francisco. Javier Arias

**Affiliations:** 1BIOFORGE Research Group (Group for Advanced Materials and Nanobiotechnology), CIBER-BBN, University of Valladolid, LUCIA Building, 47011 Valladolid, Spain; 2Recombinant Biomaterials Research Group, University of Valladolid, LUCIA Building, 47011 Valladolid, Spain; sara.escalera@uva.es (S.E.-A.); irene.alonso.sampedro@alumnos.uva.es (I.A.-S.); juan.gonzalez.valdivieso@alumnos.uva.es (J.G.-V.); arias@bio.uva.es (F.J.A.)

**Keywords:** biomaterials, aptamers, targeting, natural polymers, biomedicine, bio-targeting

## Abstract

Biomaterials science is one of the most rapidly evolving fields in biomedicine. However, although novel biomaterials have achieved well-defined goals, such as the production of devices with improved biocompatibility and mechanical properties, their development could be more ambitious. Indeed, the integration of active targeting strategies has been shown to allow spatiotemporal control of cell–material interactions, thus leading to more specific and better-performing devices. This manuscript reviews recent advances that have led to enhanced biomaterials resulting from the use of natural structural macromolecules. In this regard, several structural macromolecules have been adapted or modified using biohybrid approaches for use in both regenerative medicine and therapeutic delivery. The integration of structural and functional features and aptamer targeting, although still incipient, has already shown its ability and wide-reaching potential. In this review, we discuss aptamer-functionalized hybrid protein-based or polymeric biomaterials derived from structural macromolecules, with a focus on bioresponsive/bioactive systems.

## 1. Introduction

Biomaterials can be defined as materials that are especially suitable for intimate contact with living tissue which are selected for the development of medical devices for tissue engineering, drug delivery, and diagnostic purposes. The biomaterials in use today are derived from the metallic and polymeric ones introduced after World War II and which have subsequently been continuously and rapidly improved to perfect them, and now include both bio-inert materials, bioactive ones and even smart materials that are able to respond to stimuli and establish a biochemical dialogue with the target components of the organism [1]. The multidisciplinary interaction and combination of new knowledge from different scientific fields, especially medicine, biology, chemistry, physics, and engineering, have led to significant improvements in the advanced materials used, with subsequent and huge potential quality-of-life benefits [2].

The elementary requisites for all materials employed in biomedicine are as follows: (i) biocompatibility, (ii) appropriate function-related durability, (iii) an ability to be bioresorbed (and the degradation products must be innocuous), (iv) desired mechanical properties, (v) structural design and hierarchical control of the device structure on a nano- to macro-scale, and (vi) the availability of suitable manufacturing techniques [3,4]. One of the most demanding requirements in the fields of biomedicine and personalized medicine is currently a new approach for the creation of materials and devices that could act as biological substitutes or bio-machinery. Indeed, the need to respond to pressing demands for personalized medicine urgently requires further improvements in biomaterial science, especially in regard to the design and generation of innovative and selective products for targeting and curing different diseases (Figure 1) [5,6].

One of the most widely explored fields of advanced biomaterials for bio-targeting involves synthetic analogs of naturally derived compounds. Inspired by natural proteins present in the extracellular matrix (ECM) or structural proteins that present excellent mechanical properties, protein-derived synthetic biomaterials have been shown to be better substitutes for the more widely used metallic or chemically synthesized polymeric materials, which, although they have adequate mechanical properties, have several drawbacks in terms of their ability to be bioresorbed, their biocompatibility, and their bioactivity [7,8,9]. Different methodologies, ranging from extraction from natural sources to more modern advanced techniques, such as the recombinant production of synthetic proteins that mimic their natural counterparts, can be used to produce naturally derived materials [10]. The recombinant production of such materials, which include collagen, elastin, silk, gelatin, fibrinogen, and hyaluronic acid, amongst others, presents significant benefits in comparison with proteins extracted from animal tissue, especially their homogeneous composition, the absence of animal infectious agents, or their lower immunogenic potential [11,12]. The biodegradability of these materials allows their temporary use, when necessary, or their gradual replacement with newly formed tissue during regeneration [13]. Moreover, the design and construction of synthetic genes offers the possibility of combining unique functionalities found in nature, in a modular “Lego”-type strategy, thereby combining the complex characteristics of natural proteins with technological functionalities [14,15,16].

Naturally occurring proteins, such as collagen, fibrinogen, elastin, and silk–elastin, amongst others, have served as a source of inspiration for the biosynthesis of recombinant protein polymers, as a result of exhaustive sequence/function correlation studies subsequently selecting the protein motifs conferring the desired properties required to guarantee crucial characteristics on the new synthetic biomaterials (Figure 2).

The latest generation of materials for biomedical applications currently being studied includes precision biomaterials that are able to meet the needs of personalized medicine and to interact with the patient’s body. However, such systems require further advances in biomarker and biomaterials research in order to determine the correct target and elicit the necessary effect. To that end, tissue–material interactions must be examined, focusing on the surrounding microenvironment and temporary alterations due to the patient’s clinical state [17]. As such, a versatile biomaterial design that combines tunable building blocks for targeting, sensing, repair, and treatment with spatiotemporal control is required [18].

The inclusion of aptamers as components of multifunctional molecules, or conjugated to them, is one of the most promising approaches for determining biomarkers and obtaining advanced devices. Aptamers are small single-stranded nucleic acids with a specific affinity for a target; as such, they can be considered to be analogues of antibodies but can be produced in a more economical and easy manner. Moreover, aptamers are neither immunogenic nor toxic, both of which are essential properties for a biomedical device. The design and screening of aptamers for the targeting of personalized biomarkers has led to innovative strategies for previously unresolved clinical issues [19].

Herein we describe the improvements obtained upon conjugating naturally derived biomaterials with aptamer target systems (Table 1). Due to their origin, each of these biomaterials presents specific and distinct physiological and mechanical characteristics, thereby offering various alternatives to adapt to and satisfy specific requirements, while aptamer conjugation confers the required specificity.

## 2. Biomaterials

### 2.1. Collagen and Gelatin

Collagen is the most abundant protein in mammalian fibrous tissues, playing a key role in the skin, bones, and connective tissues. It is mainly produced by fibroblast cells and provides mechanical support and resistance to plastic deformation [49]. At least 29 types of collagen have been reported (I–XXIX), with collagen type I, which accounts for 90% of all collagen in the human body, being the most abundant [50]. Collagen has four levels of structural organization, with the primary structure comprising the tripeptide sequence -(Gly–X–Y)*_n_*- where Gly is glycine and X and Y are generally proline and hydroxyproline, respectively. The secondary structure is a chain of triplet repetitions; the tertiary structure is a stable triple helix, and the intramolecular or intermolecular crosslinking of the insoluble fibers entails the quaternary structure of collagen [51].

Collagen-based biomaterials have great potential for biomedical applications due to their easy processing, biodegradability, desired biocompatibility, low immune response, absorption in the body, etc. [51]. Although natural collagen derived from animal tissue has been widely used for drug delivery [52] or tissue engineering [53], its quaternary structure and insolubility limit its processing for the production of moldable scaffolds. In addition, its limited mechanical strength and potential antigenicity [54] have led to the need to treat collagen enzymatically or by varying the pH and temperature, thus forming atelocollagen or gelatin.

Atelocollagen is a collagen derivative obtained upon proteolytic treatment of collagen type I. This treatment eliminates the telopeptides—sequences located at the N- and C-termini of collagen—responsible for collagen interchain crosslinking and initiating correct fibrillogenesis. As such, the removal of telopeptides entails an increase in solubility [55]. As these telopeptides do not possess the repeating Gly–X–Y motif, they have a non-helical conformation [56]. In addition, they have been reported to be responsible for collagen immunogenicity [57]. As such, atelocollagen exhibits higher biocompatibility and biodegradability than collagen and is therefore used in numerous biomedical and cosmetic applications.

The acidic or basic treatment of collagen gives rise to gelatin, which is a natural polymer obtained after denaturalization of the triple helix in collagen. There are two types of gelatin: Type A is obtained after acid treatment of collagen, and type B is obtained after basic hydrolysis. The difference between them is the isoelectric point (IEP) of the resulting gelatin: Acidic gelatin possesses an IEP of about 5, similar to collagen, whereas basic gelatin has a higher IEP, of about 9 [58]. Another key property is its thermo-responsiveness as gelatin undergoes a gel-to-solution transition at around human body temperature to form physical thermo-reversible gels when cooled. As such, gelatin hydrogels have poor mechanical strength, weak elasticity, and low shape stability, and they usually need crosslinking agents to overcome these limitations for use in biomedical applications [59,60]. Gelatin has also been widely used as a scaffold for tissue engineering and regenerative medicine because of its cell-stimulatory properties.

Although their origin is the same, collagen and its derivatives have different properties that broaden the field of application for these materials for different biomedical applications, such as cancer therapy, wound healing, or aptasensor development, which are reviewed in this section.

#### 2.1.1. Cancer Therapy

Some of the most innovative approaches for cancer therapy include the use of post-transcriptional gene silencing (PTGS) or RNA interference (RNAi). RNAi is a very powerful technique that allows the suppression of genes with high specificity [61]. This natural mechanism has been employed as strategy to selectively silence, or deactivate, genes involved in tumorigenesis. It employs small interfering RNAs (siRNAs), which are molecules that can suppress the expression of certain genes by degrading the messenger RNA (mRNA) after transcription. Many studies to date have successfully used RNAi to control aberrant cell proliferation and cancer. RNAi requires a suitable delivery system to both protect the RNA and transport it to the target tissue, thereby overcoming the main obstacles of this technique, namely the fact that the low molecular weight of siRNA and its hydrophobicity can lead to diffusion through cell membranes [62]. In this regard, atelocollagen is a promising carrier for siRNA that has been reported to increase cellular uptake and promote the release of genes and oligonucleotides [63].

Common tumors, such as those of the prostate, breast, and lung, frequently metastasize to the bone, with the bulk of the tumor often being found in the bone at the time of death [64]. Hao et al. have developed a system for treating prostate-tumor-derived bone metastasis, using atelocollagen-mediated siRNA delivery [20] in which atelocollagen is chemically conjugated with a prostate specific membrane antigen (PSMA)-targeting aptamer. PSMA is highly overexpressed in prostate cancer [65], and ligands that bind to PSMA are endocytosed by clathrin-coated pits [66], meaning that a PSMA-targeting approach is a good choice for both targeting metastatic cells derived from prostate cancer and their uptake. The atelocollagen–aptamer complex was used as a carrier for miR-15a and miR-16a, two silencing microRNAs known to inhibit cell proliferation and promote the apoptosis of cancer cells [67]. The efficacy of this delivery system, known as miRNA/ATE–APT, was tested both in vitro, with a higher transfection being found in complexes with aptamer, and in vivo. For the in vivo experiments, a murine model of prostate cancer bone metastasis was used, and after treatment with the miRNA/ATE–APT complex, survival was found to increase significantly, compared to complexes lacking the aptamer or controls. Atelocollagen is therefore both a suitable miRNA delivery system and a biocompatible and degradable scaffold for cancer therapy.

Another device employed for RNAi involves the gelatin–silica nanogels (GSNGs) described by Zhao et al. In an initial study, gelatin–siloxane nanoparticles were produced by using a sol–gel method and decorated with TAT cell-penetrating peptide, as a novel shuttle for gene therapy [68]. These nanoparticles had a size of 200 nm and a positive surface charge and were found to enhance gene transfection efficiency when compared with commercial reagents, thus also meaning that they are suitable for the delivery of nitric oxide for vascular cell regulation and may be a promising agent for the control of restenosis [69].

In a subsequent study, the same authors developed redox-sensitive nanogels with nucleolin-targeted AS1411 aptamer, HA2 fusogenic peptide, and poly(ethylene glycol) (PEG) [21] (referred to as GS-PEG/HA2–Apt). AS1411 (also known as AGRO100) is a DNA aptamer with a G-quadruplex structure that can bind to nucleolin discovered by Bates et al. [70]. It was the first aptamer tested in humans for the treatment of cancer [71]. Nucleolin is involved in a wide range of cellular procedures, although its expression and localization in rapidly proliferating cells is abnormal, which results in a higher malignancy in cancers, thus making it a good candidate for targeted cancer therapy [72]. The addition of a hydrophilic polymer, such as PEG, to the system reduces the reticuloendothelial system uptake, thus allowing a longer circulation time for the resulting devices [73]. The addition of fusion peptides, such as HA2, has been reported to increase cellular internalization and transfection efficiency [74]. These authors optimized in vitro assays to determine the transfection efficiency with A549 cancer cells and normal 3T3 fibroblasts. The cellular uptake in A549 cells was five- to tenfold higher for GS-PEG–Apt and GS-PEG/HA2–Apt compared to 3T3 cells. In addition, AGRO100 and HA2 showed a synergistic effect in transfection efficiency. These authors also demonstrated the transfection efficiency of GSNGs in vivo by intravenous administration of decorated nanodevices to A549 tumor-bearing mice. The accumulation of GS-PEG–Apt and GS-PEG/HA2–Apt was found to be significantly higher in tumors compared to naked GS-PEG. However, in vivo luciferase expression quantification demonstrated that transfection efficiency did not correlate with in vivo biodistribution. Thus, while the highest accumulation was found in tumors and liver, gene expression was higher in heart. Furthermore, the GS-PEG/HA2–Apt nanogel could be leveraged for gene therapy in heart diseases, since gene expression was highest in heart tissue with lower accumulation.

A further improvement for this work was the addition of the stimuli-responsive release of siRNA under reducing conditions to the nucleolin-targeting via AS1411 aptamer [22] (Figure 3). Nanocarriers that respond to changes in redox potential or pH have been reported to enhance drug delivery at targeted sites and to reduce the leakage in blood circulation [75,76]. The equilibrium between glutathione (GSH) and glutathione disulfide (GSSG) in a normal tissue microenvironment is necessary to maintain cellular redox homeostasis, which allows for correct cell growth and function [77], whereas GSH is overexpressed in the cytoplasm of cancerous cells [78]. Modification of the siRNA with deoxynucleotides has been reported to maintain the stability of the sense strand but decrease the RNAi efficacy to 40% [79]. In this study, siRNA was conjugated to active Apt–GSNGs by way of a disulfide-crosslinking reaction, meaning that, in the presence of GSH, the siRNA is released by disulfide cleavage and allowing selective release only in cancerous cells. The ability of the device to silence genes was examined in vitro, using the luciferase reporter gene system. After 48 h, the gene knockdown efficiency in nucleolin-overexpressing cells was fivefold higher than for non-cancerous cells, thus proving the specificity of the delivery system and highlighting its potential for cancer therapy.

#### 2.1.2. Wound Healing

One of the most interesting properties of aptamers is their ability to hybridize with their complementary DNA sequences, thus preventing them from binding to target molecules. This “antidote” oligonucleotide inactivates the aptamer, provoking aptamer–protein dissociation and releasing the previously bound molecule [80]. In this regard, Soontornworajit et al. developed a system based on complementary oligonucleotides (CO) that could trigger the release of platelet derived growth factor (PDGF-BB) when added to the sample [23]. This system is a composite of anti-PDGF-BB aptamer-tethered particles and a gelatin matrix. Release experiments were carried out and the aptamer–gelatin composite was found to release 5.3% of the PDGF-BB retained, in contrast to 47.9% released from the control. The addition of CO solutions with increasing concentrations resulted in little release in the first hour, although the amount of protein in the medium increased abruptly and proportionally to the CO concentration 24 h after release. This work is a proof-of-concept for a triggered release system that could be used in many biomedical applications, ranging from growth factor release to anticancer therapy or tissue engineering.

The most common hydrogels used as scaffolds for regenerative medicine often lack the bioactivity required for cell integration and proliferation. To overcome this issue, the conjugation of different bioactive molecules to the synthetic matrix is a common solution. For example, the Wang group developed a macroporous gelatin–PEG hydrogel with the ability to retain bioactive molecules and cells [24]. These hydrogels were prepared by free radical polymerization of PEG, gelatin–methacryloyl, and anti-VEGF aptamers coupled with gas formation [81]. VEGF sequestration and release were studied, and it was found that, when the macroporous hydrogels were conjugated with the aptamer, the sequestering efficiency reached 91.6%, compared to 5% in the absence of aptamer. Moreover, the release rate was dramatically affected by the aptamer-functionalized hydrogels, with values of 89.6% and 94.7% at days 1 and 2, respectively, for the naked hydrogel and 4.2% and 7.7% for the aptamer-containing hydrogel. A tube-formation assay using human umbilical vein endothelial cells (HUVECs) was carried out to evaluate whether the released VEGF was functional, as it plays a leading role in angiogenesis [82], with longer tubes indicating a higher bioactivity. Media collected from aptamer hydrogels at days 7 and 14 stimulated 73% and 43% of cells, respectively, to form tubes, with no significant angiogenesis found in cells treated with naked hydrogels. This is consistent with the 95% release observed at day 2 for these hydrogels. Finally, the hydrogel with cells and VEGF was evaluated under serum-reduced conditions, and it was found that cells could survive for up to 14 days, proving its ability to mimic the ECM. Future work with this system will include in vivo testing of the device.

Another unsolved problem in bioactive material development is the easy loss of bioactivity when stored [83]. To overcome this, the same group developed a gelatin-based material that can transform from a solid to a hydrogel upon hydration and which can sequester proteins [25]. Briefly, the material was synthesized in a two-step desolvation process [84] and then chemically crosslinked. This process begins with the dissolution of gelatin in water and subsequent precipitation with acetone. The precipitated gelatin is then re-dissolved and cooled to room temperature, and then the pH is adjusted to 2. Dropwise addition of acetone to the solution precipitates the gelatin nanoparticles (GNPs), and the subsequent addition of glutaraldehyde allows crosslinking of the GNPs. The nanoparticle suspension is then lyophilized, to obtain sponge-like structures formed by physical nanoparticle assembly during lyophilization process. For the anti-VEGF aptamer GNPs, conjugation was performed prior to lyophilization. In the tube-formation experiment, the VEGF released from the aptamer–hydrogel was found to be able to stimulate angiogenesis to the same extent as commercial VEGF, whereas the naked hydrogel showed low stimulation, probably as it had released nearly all the VEGF by day 1.

According to previous studies demonstrating the ability of gelatin to stimulate cell adhesion [85,86], the authors studied the synergistic effect of the gelatin scaffold and VEGF release. HUVECs showed no difference in viability between the control and functionalized hydrogels, which contrasts with the data available in the literature. The authors suggest that this contradiction may be due to the rigidity of the biomaterial compared to those used in other studies. As growth factor release is crucial for wound healing, the devices were implanted into a dorsal skin wound in mice to check the ability of the released VEGF to promote angiogenesis. A subsequent histological study showed that there was no significant difference in angiogenesis between the naked and aptamer-conjugated GNPs, which does not fit with the in vitro data. The authors concluded that this inconsistency may be due to the short time period chosen to perform the in vivo experiment (10 days), in addition to the fact that in vivo release may be much slower than in vitro; therefore, a further in vivo experiment over a longer time period may show much more enhanced angiogenesis.

#### 2.1.3. Biosensing Application: Current Advances and Progress of Using Natural Protein-Based Polymers

Given their ability to bind ligands with high specificity and affinity, aptamers are starting to carve out a niche in biosensors field. Biosensors with an aptamer as recognition element are known as aptasensors, and biosensor development is one of the main fields in which collagen and its derivatives play a leading role, because they provide a more biocompatible matrix for immobilization of the aptamers. Collagen-based aptasensors have been developed for the detection of different molecules, including thrombin [64,65], dopamine [28], or chloramphenicol [29].

Derkus et al. developed an aptasensor for thrombin detection using jellyfish collagen as matrix [26]. Although jellyfish is not the most common source for collagen, it has the potential to become one of the main such sources, as it has a high collagen content and fewer potential side effects than other animal sources (i.e., bovine spongiform encephalopathy or avian influenza) [87]. The fast and accurate detection of thrombin in biological samples is crucial for the medical field because this substance is involved in different diseases such as Alzheimer’s, multiple sclerosis or ischemia [88]. Thrombin is also involved in blood clot formation as it degrades fibrinogen into fibrin [89].

Although several articles concerning sensors for thrombin detection have been reported, the work from Derkus et al. was the first to involve a pre-clinical study. The carbon electrodes of this sensor were biofunctionalized with collagen, and, subsequently, amine-modified thrombin aptamer was immobilized on the surface via glutaraldehyde crosslinking reaction. Thrombin was detected by electrochemical impedance spectroscopy with samples derived from patients, and a limit of detection of 6.25 nM thrombin was achieved. This aptasensor sensitivity was improved by the addition of egg-shell-derived hydroxyapatite nanoparticles to jellyfish collagen, thus forming a nanocomposite to coat the sensor, with the detection limit decreasing from 6.25 nM thrombin to 0.25 nM [27].

Wei et al. developed a sensor for dopamine detection based on a collagen–graphene oxide composite [28]. Dopamine is a neurotransmitter that plays a role in movement regulation in the brain. Parkinson’s or Alzheimer’s diseases, among others, are related to a deficient metabolism of dopamine [90,91]; therefore, its detection is of particular importance for the diagnosis of these diseases. In this sensor, a dopamine-binding aptamer was embedded in the collagen matrix that covered the graphene oxide. Collagen conferred biocompatibility on the graphene oxide and was found to also increase the current response. The aptasensor developed was used to detect dopamine with a detection limit of 0.75 nM.

Hamidi-Asl et al. developed an aptasensor for chloramphenicol (CAP) sensing that incorporated a protective gelatin matrix [29]. Chloramphenicol is a broad-spectrum antibiotic commonly used in animals, and its detection in products for human consumption is therefore very important to avoid allergic reactions or bacterial resistance [92]. To develop this biosensor, a thiolated single strand DNA (ssDNA) aptamer for CAP detection was mixed with gelatin and attached to the gold surface of the electrode by Au-S bonding. When CAP is detected, the aptamer changes its conformation to bind it, thus facilitating electron exchange between the target and the electrode. Differential pulse voltammetry was chosen as the sensing technique, to investigate the effect of gelatin on the efficiency of the aptasensor. Thus, a gelatin B (GelB)-modified electrode showed a higher current signal than its gelatin A (GelA) counterpart, and when the aptamer was incorporated, GelA showed no improvement in efficiency but GelB showed a good response. As such, the hydrophilic domains of hydrated GelB provide a suitable environment for aptamer immobilization, while maintaining its native configuration. Furthermore, the negative charge of GelB at physiological pH interacts electrostatically with the CAP, which has a p*K*_a_ of 11.03 [93] and is positively charged at that pH. This interaction helps the accumulation of CAP molecules on the surface of the electrode. GelA, in contrast, is positively charged at physiological pH, which may explain the lack of effect on the sensor’s efficiency. The detection limit for this aptasensor was 0.183 nM dopamine, which is nearly 10-fold higher than for the aptasensor lacking the protective gelatin matrix developed by the same group, which had a detection limit of 1.6 nM [94].

### 2.2. Elastin

Elastin is an elastomeric protein and one of the most abundant components in the extracellular matrix (ECM) of vertebrate tissues, along with collagen and glycosaminoglycans [95]. The function of elastin is to bestow stability and resilience to a wide range of tissues and organs (such as the skin, lungs, or blood vessels), as well as to contribute to cell signaling [96]. Elasticity arises due to the ability of this protein to be deformed reversibly and repeatedly without losing energy or its mechanical properties [97]. Although elastin is a hydrophobic insoluble network, its monomeric precursor, tropoelastin, is a soluble protein of approximately 60 kDa. The sequence of tropoelastin comprises several repeated hydrophobic domains, which are responsible for its elastic properties, along with hydrophilic crosslinking domains. When tropoelastin is secreted in the ECM, it undergoes covalent crosslinking in the presence of lysyl oxidase (LOX) [98], to form an insoluble complex and fibrillar structures [99].

The development of elastin-based scaffolds began with isolation of the protein from elastic animal tissues. However, current recombinant DNA technology has allowed the synthesis of synthetic analogues. In the 1970s, Urry et al. published the development of a family of polymers inspired by the composition of elastin, known as elastin-like polypeptides (ELPs) [100], and in the early 1990s, the methodology for their recombinant production was developed in order to produce elastin-like recombinamers (ELRs) [101]. Recombinant protein polymers have a remarkable versatility, high intrinsic biocompatibility, and self-assembling properties, and they can be used to construct modular building blocks that introduce interesting and novel structural or functional motifs that mimic the properties of complex proteins [102]. All of these elastin-derived polymers possess a “backbone” involving the repetition of certain motifs which take part as elastin building blocks. The most widely studied motifs are derived from elastomeric hydrophobic domains and include the (VPGXG) pentapeptide, a conserved and repeated motif in which the fourth amino acid (X) can be any of the 20 natural amino acids except L-proline [103]. Substitution of the fourth amino acid in (VPGXG) generates changes in perhaps the most interesting property of ELRs, namely their thermo-responsiveness, which produces a thermodynamically driven reversible phase transition [104]. Depending on the ELR composition, several controlled aggregation phenomena occur in response to temperature changes [105].

Due to their high versatility and special biochemical characteristics, synthetic biomaterials derived from elastin have been a field of interest in recent years [106,107,108], mainly in the biomedical field. Thus, the possibility to combine different structural protein domains or include bioactive amino acid sequences has opened the door to the use of ELRs as drug- [109] and gene-delivery systems [110], bio-inks for 3D printing [111], and biomaterials for tissue engineering and regenerative medicine [112,113].

#### 2.2.1. Cancer Therapy

In particular, gene therapy is an innovative approach to provide alternative targeted therapies that allow the replacement of a gene responsible for a disease, stimulate an immune response or induce suicide genes that cause tumor cell death. Thus, suicide gene therapy could be used to eradicate tumors with no undesired damage to healthy tissues [114,115]. Some potential gene-therapy treatments have been validated experimentally and have recently entered clinic trials [116], with up to 70% of these being related to cancer and requiring an appropriate vehicle system or vector to protect and target them correctly [117,118]. In this scenario, nanodevices have found an important role because they are able to protect drugs in the bloodstream while limiting side effects and enhancing proper biodistribution. Moreover, different surface properties or aptamer-conjugation can provide the nanoparticles with selective targeting [119,120,121].

Over the past six years, numerous innovative studies related to the design of specific gene-delivery systems for breast cancer, based on aptamer–ELR–pDNA devices for suicide gene therapy, have been published. This type of cancer is the most common cancer diagnosed in women [122], thus highlighting the promising potential of this therapeutic approach for human society. However, current treatments are not cancer-specific and can cause several side effects [123]. As such, advanced anticancer therapies are required and numerous efforts have focused on the development of targeted biological strategies to maximize efficacy, thereby reducing these side effects.

Several studies have used non-viral systems to transport therapeutic plasmids to breast tumors using specific promoters. Thus, in 2016 Piña et al. reported a strategy based on the use of a polycationic ELR decorated with MUC1-specific aptamers as a vector for the specific delivery of therapeutic material into MUC1-positive (MUC1+) breast cancer cells [30] (Figure 4). This work was based on previous in vitro studies in which the authors obtained promising results by using polycationic lysin-based ELRs fused to bioactive domains as gene-delivery systems [124]. The positive charge of the ELR allowed complexion with the negatively charged plasmid DNA (pDNA) via electrostatic interactions to obtain a polyplex with a particle size of 140 ± 20 nm, which is appropriate to be internalized by endocytosis (<200 nm) [125,126].

Once a stable and positive ELR–pDNA system had been achieved, the MUC1-targeted aptamer 5TR1 was covalently conjugated to the polyplex. The target of this aptamer is the aberrant under-glycosylated form of transmembrane glycoprotein MUC1. This glycoprotein is overexpressed in its under-glycosylated form in several cancers, such as breast cancer, and entails a poor prognosis [127].

Previous studies showed that positively charged ELRs were innocuous when incubated with blood components [124]. As such, the cytotoxicity of control polyplexes bearing luciferase as a reporter gene was determined in different human cell types, namely a breast cancer line overexpressing aberrant MUC1 (MCF-7), a non-overexpressing MUC1 liver carcinoma cell line (HepG2), and three primary cell types (endothelial, mesenchymal, and fibroblasts). The results demonstrated the excellent biocompatibility of polyplexes and an increase in the transfection rate of the MUC-1 presenting MCF7 cells due to the specificity provided by the 5TR1 aptamer, in contrast to the rest of the cell lines. The specificity of the aptamer–ligand interaction and cellular uptake in comparison with nude polyplex behavior were proven in vitro by way of a competitive inhibition assay in the presence of anti-MUC1 antibody. Flow cytometry and fluorescence microscopy enabled cellular internalization and nuclear localization of the ELR–pDNA–5TR1 system in MUC1+ MCF-7 cells to be observed. Incubation of cell cultures with the ELR–5TR1 system loaded with the therapeutic pDNA (pCMV–PAP) resulted in an almost 95% increase in the death rate for breast cancer cells, thereby suggesting an efficient expression of functional PAP-S toxin in the transfected cells. These results, together with those obtained in human primary cells, highlight the selective effect of the transfected plasmid in MUC1-presenting breast cancer cells, which opens up a route to potential in vivo therapies.

These innovative in vivo therapies were further developed by the same group in a more recent work [31] in which the authors described a double-lock device comprising the polyplex ELR–PEG–5TR1 loaded with therapeutic DNA, with which an increase in the potential harm with no damage to non-tumor cells could be achieved both in vitro and in vivo. The therapeutic agent designed for this research was pDhMUC1–ricin, which comprises the hMUC1 promoter, a MUC1 tumor-specific promoter, preprotrypsin leader sequence, and ricin gene. The first lock is the specificity of the 5TR1 aptamer for breast cancer cells, while the second lock ensures expression of the cytotoxic drug ricin in MUC1 overexpressing cells only. Indeed, the therapeutic DNA gene is under control of the hMUC1 promoter, while the preprotrypsin leader guarantees over-synthesis and ensures secretion of cytotoxic ricin to obtain a ripple effect in adjacent cells [31].

The in vivo cancer model was obtained by subdermal injection of xenografted human cancer cells into athymic mice, which were treated with aptamer-decorated polyplexes containing the gene encoding for ricin.In contrast to the placebo group, treatment significantly decreased both tumor volume and weight. The findings also suggested a dose-dependence of the therapeutic effect, which was confirmed by immunohistochemical assays. Thus, evaluation of the cancer-prognostic MUC1 protein and the proliferative marker Ki-67 showed that treatment acts in a specific and selective way against tumor cells, with no side effects in healthy ones. In light of all of these results, Piña et al. were able to develop an efficient nanodevice for breast cancer gene therapy [31].

In 2019, Mie et al. engineered nanoparticles (NPs) for specific loading of the anticancer drug paclitaxel (PTX) [32]. These NPs consisted of ELRs with a poly(aspartic acid) tail (ELR-D system), which were functionalized with the MUC1-specific aptamer S2.2. This aptamer is also able to bind to the extracellular domain of the overexpressing MUC1+ in breast cancer cells. To that end, the ELR-D system was fused to the C-terminus with the Gene A* protein sequence. Gene A* protein plays a role in replication of the ϕX174 bacteriophage, and is able to retain DNA strand-transfer reactions by cleaving the recognition sequence. A phosphotyrosine diester is formed as a result of the enzymatic binding between the aptamer and the cleavage site in the sequence. According to the DLS results, electrostatic interactions between the poly(aspartic acid) tail and Gene A* hindered the formation of appropriate size aggregates. This problem was resolved by mixing ELP-D–Gene A* and ELP-D with no Gene A* protein, which allowed the production of nanoparticles with a size of around 30 nm. In vitro assays were performed with PTX encapsulated within nanoparticles. MCF-7 cells were cultured with PTX-loaded nanoparticles, showing a high cytotoxic effect after three days, in contrast with free PTX. This result therefore suggested cellular internalization of the nanoparticles, although the results obtained should be interpreted in the light of further experiments to clarify the mechanism of action.

Guo et al. tried to overcome problems derived from the conjugation of ELR with DNA aptamer (S2.2) in their previous work [32]. In this subsequent study, the ELR sequence was designed without a poly(aspartic acid) tail. Furthermore, the catalytic domain of Porcin Circovirus type 2 replication initiation protein (pRep) was selected as a linker for the ELR and aptamers to reduce the nanoparticle size due to its lower molecular weight. DLS showed the ability of the ELR–pRep system to form 40 nm nanoparticles with the DNA aptamer enzymatically conjugated at their surface [33]. MUC1+ MCF-7 cells were incubated with MUC1 aptamer-displaying ELR nanoparticles carrying encapsulated PTX for in vitro experiments, and cell viability assays demonstrated that the NPs successfully delivered the PTX drug to cancer cells. Nevertheless, the absence of negative controls does not allow the specificity of the designed delivery system for MUC1+ cancer cells to be confirmed.

#### 2.2.2. Other Applications

In addition to the previously described approaches, applications of ELR–aptamer conjugates grow every day. One example is the work reported by Vogele et al., who have designed amphiphilic ELR-based vesicles that can accommodate a cell-free gene expression system. To prove the efficacy of transcription, correct transcription of a fluorogenic dBroccoli RNA aptamer inside the vesicles was monitored by fluorescence measurements. In addition, the expression of a fluorescent protein proved successful compartmentalization. The authors hypothesized that membrane peptides could be expressed inside the vesicles, which was subsequently verified by FRET assays. The results showed a 4.4-fold increase in vesicle growth because of internal ELR-membrane expression and its inclusion into the vesicle membrane [34].

This work was continued by Frank et al., who developed giant vesicles based on amphiphilic elastin-like recombinamers that act as synthetic cells which are able to develop and express genes at the same time [35].

### 2.3. Fibrinogen and Fibrin

Fibrinogen is a fibrous glycoprotein of 340 kDa that comprises 2Aα, 2Bβ and 2Ɣ polypeptide chains linked by disulfide bonds. It is a plasmatic protein whose plasma concentration can increase from 2.5 to more than 7 mg/mL during the acute phase of inflammation [128]. Moreover, although it is a soluble molecule, it converts to insoluble fibrin catalyzed by thrombin, a serine protease that is activated after a cascade of reactions triggered by injury to promote coagulation [89].

The cleavage of fibrinogen chains determines both fibrin polymerization and the release of fibrinopeptides. Free fibrinopeptides are bioactive and chemotactic and play a role in tissue repair [129]. Fibrin subsequently assembles into protofibrils, the aggregation of which leads to a clot. In vivo, there is a dynamic equilibrium between clotting and fibrinolysis, such that the conversion of fibrinogen to fibrin and dissolution of the clot must be carefully regulated [130].

The fibrin network provides both physical support for cell infiltration, which helps to rebuild the damaged tissue, and a rich wealth of signals to direct cell behaviors after injury because of its numerous binding sites for growth factors or integrins [131].

Fibrin has been widely employed as a biomaterial because of its important role in hemostasis, directing tissue regeneration after injury, ease of tunability, and its ease of polymerization [132]. Moreover, fibrin-based biomaterials are bioactive and easy to manipulate. This manipulation can lead to the formation of different scaffolds with optimized characteristics, such as different porosity, fiber thickness, and degree of branching of the polymerized gel [133], which can tune its mechanical properties, thus making fibrin a very versatile material for a wide range of biomedical applications from tissue engineering and wound repair to stem cell delivery or angiogenesis. In addition, the possibility of making fibrin scaffolds after the extraction of thrombin and fibrinogen from patients’ blood makes it an autologous and cheap material [134].

Fibrin has been used as a scaffold for the engineering of diverse tissues for decades. The most widely used forms of fibrin scaffolds are fibrin hydrogels, fibrin glue, and fibrin microbeads (FMBs) [135]. Some of the advantages of fibrin hydrogels as biomaterial are their low immunogenicity, injectability to form implants in situ, and tunable degradation; meanwhile, the main disadvantages relate to the mechanical performance of the material in terms of shrinkage and low stiffness [136].

#### 2.3.1. Tissue Regeneration

Hydrogels provide a suitable scaffold for biomedical applications, as they are able to supply mechanical support and entrap drugs, cells, or biological macromolecules, such as growth factors (GFs), for controlled delivery. GFs are soluble proteins secreted by cells that are responsible for regulating several biological processes, such as cell proliferation, migration, or differentiation, which all play a pivotal role in tissue regeneration [137]. Some of the most widely studied GFs are those involved in angiogenesis, such as vascular endothelial growth factor (VEGF) and fibroblast growth factor-2 (FGF-2) [138], or platelet-derived growth factor-BB (PDGF-BB) for the enhancement of tissue formation [139]. These GFs hold great potential as therapeutic agents, despite having only a short half-life. To overcome the rapid degradation of GFs, one strategy that has proven effective is to sequester them in hydrogels, thus allowing a spatiotemporal control of their release. Aptamer-functionalized hydrogels are excellent candidates for the sequestration of GFs because of their ability to retain them and regulate their release rates, as well as their ability to provide a protective environment in which they can be embedded. A further advantage of this approach is the possibility of sequestering endogenous GFs, thus preventing them from triggering an immunogenic response.

The Wang group has developed several systems for this purpose [37,38,39,40]. Thus, they initially developed fibrinogen–aptamer macromers (Ap–Fgs) that gave rise to fibrin hydrogels with a molecular recognition capability, using an anti-VEGF aptamer for wound-healing applications [37]. The Ap-Fgs were synthesized using a thiol-ene reaction between acrylate-modified fibrinogen and SH-modified aptamer, which has been reported to provide a stable and strong crosslink in a biocompatible manner for subsequent bioconjugation of nanomaterials [140]. Upon addition of thrombin, a hydrogel with the ability to specifically embed VEGF was formed. This VEGF was released in a sustained way while maintaining its bioactivity. In the native hydrogel 97% of VEGF was released in the first three days. On the contrary, hydrogels with aptamer released 35.7% of VEGF in the first hours, followed by a daily release rate of 3–4% between days 2 and 14. The functionality of the hydrogels was determined in vitro in different cell types, such as keratinocytes, fibroblasts and endothelial cells. Keratinocytes grew slightly faster on the VEGF-loaded Ap-Fn than on the controls, fibroblasts grew similarly in both VEGF-loaded Ap-Fn and controls, and HUVECs were able to grow in all hydrogels but, like keratinocytes, did so faster in VEGF-loaded Ap-Fn hydrogels. To check if growth occurred as a result of VEGF release, these authors measured the expression of phosphorylated VEGF receptor 2 (p-VEGFR-2), which is involved in VEGF-induced cell proliferation [141], finding that cell showed a higher expression, thus confirming the action of VEGF release. These results show that the different cell types involved in wound healing can survive and proliferate on the VEGF-loaded Ap-Fn. Moreover, the endothelial cell cultures showed a specific response to the GF. For in vivo experiments, the gels were implanted in dorsal skin wounds of mouse models for full-thickness skin wound healing. By day 7, VEGF-loaded Ap-Fn hydrogel treated wounds were significantly smaller with respect to the controls, and by day 13, their size was 7.2 ± 3.2% of the starting size. Histology revealed a shorter distance between wound margins, and anti-CD31 staining confirmed the proangiogenic effect, showing 25% more blood vessels in the VEGF-loaded Ap-Fn hydrogel compared with controls. These findings confirmed that sustained VEGF release induces both angiogenesis and tissue regeneration.

This system can be improved by implementing the co-delivery of different growth factors. To that end, the same group functionalized the previously described hydrogels with both anti-VEGF and anti-PDGF-BB aptamers [38]. As VEGF plays an initial role in angiogenesis and PDGF-BB is more important for subsequent stabilization of the blood vessels [142], authors have also developed several aptamers with decreasing affinities that could allow a sequential release [143]. These aptamers were synthesized by using the Systematic Evolution of Ligands by Exponential Enrichment (SELEX) technique [144]. The continuous screening process of SELEX leads to DNA molecules with high specificities and affinities [145]. To achieve aptamers with different degrees of specificity, the sequence of the most specific aptamer was muted by introducing nucleotide substitutions that lead to lower affinity or different secondary structures. Affinity screening of the derived aptamers was carried out by surface plasmon resonance, and the aptamers were chosen based on the affinity with which they needed to be retained. A VEGF aptamer with lower affinity and a higher affinity aptamer for PDGF-BB were selected because PDGF-BB has to be retained and subsequently released. On this occasion, the hydrogels were formed in situ by subcutaneous injection to check the angiogenesis in vivo. Blood vessel quantification revealed that co-delivery of VEGF and PDGF-BB was able to promote the formation of mature and stable blood vessels, thus demonstrating a marked potential for regenerative purposes because of the synergic effect of both growth factors. This opens up a route to a new level of growth factor delivery in which aptamers that trap different growth factors can be used in the same device, thus allowing faster and more accurate tissue regeneration.

After their success in wound healing, engineered fibrin hydrogels for the release of VEGF were also studied for bone regeneration purposes [39]. Thus, Juhl et al. assayed bone regeneration promoted by the implant of different VEGF-loaded hydrogels with and without anti-VEGF aptamer in mice with critically sized calvarial bone defects. VEGF concentrations of 2.5 and 10 µg/mL were chosen for the in vivo assay based on previous literature values of 2.5 µg/mL for skin defect assays [37] and 10 µg/mL for femoral fracture models [146]. In general, all of the VEGF-loaded aptamer hydrogels showed an increase in vascularization compared to the hydrogels without aptamer at 21 days post-injury. Bone healing was only noticeable for the highest VEGF concentration at 21 days, probably because a longer time is required to regenerate bone tissue. A good in vitro model for bone healing could have reduced the number of conditions tested for this in vivo model. Future work with these hydrogels could include the inclusion of specific bone regeneration GFs, such as bone morphogenetic protein 2 (BMP-2), or PDGF-BB in the device, or co-delivery of different growth factors with synergic effects, a technique that this group previously tested with success in wound healing [38].

#### 2.3.2. Cancer Therapy

In a more recent study, the Wang group has employed aptamer-functionalized fibrin hydrogels to enhance the survival of mesenchymal stem cells (MSCs) used for regenerative purposes and cancer therapy [40]. MSCs are multipotent cells, possess anti-inflammatory and immunomodulatory activities, and can proliferate and differentiate into important cell lineages [147]. As such, they are one of the best cell choices for tissue regeneration [148]. MSCs have also been studied in the treatment of different cancers, although this has been a controversial topic because of their divergent roles in both tumor development and suppression [149].

Although MSC-based anticancer therapies often require controlled delivery, unfortunately, MSCs show low survival rates after release in vivo. To overcome these issues, more effective ways of delivery, such as polymer-based scaffolds, genetic manipulation, preconditioning before transplantation, or combined administration with other molecules or cells, have been studied [150]. The 3D culture of MSCs in spheroids is one technique that has been shown to enhance cell survival with regard to the delivery of isolated cells [151]. To improve this approach, MSC spheroids were embedded within anti-VEGF and anti-PDFG-BB aptamer–fibrin hydrogels, thus providing a three-dimensional scaffold that could better mimic cell–cell interactions in real tissue. The hydrogel provides a scaffold and allows the interaction of MSCs with the damaged tissue upon degradation. The release of PDGF-BB was also shown to result in higher survival of MSCs upon stress. Hydrogels were tested by subcutaneous delivery in the dorsal region of mice, and the promotion of MSC survival in vivo was achieved. Unfortunately, the angiogenesis promoted by the release of VEGF and PDGF-BB was not satisfactory, perhaps due to the brevity of the experiment.

Fibrin could be an interesting biomaterial for use as a scaffold for the development of reliable tumor models, as fibrin clots resulting from blood vessel destruction are closely related to cancer invasion and metastasis [152,153], and fibrin is also an important component of the tumoral stroma [154].

Fujita et al. developed the “selective oligonucleotide entrapment in fibrin polymers” (SOEF) approach, which allows the incorporation of several amphiphilic groups into fibrin gels via chemically modified thrombin-binding DNA aptamer (TBA) [41] (Figure 5). These modifications in TBAs confer an amphiphilic nature and nuclease resistance on the oligonucleotides [155,156].

In contrast to the hydrogels developed previously by Zhao et al. [37], in this case the aptamer selectively binds to thrombin instead of fibrinogen, thus providing an alternative means of functionalizing the hydrogel. To analyze the new functionality incorporated into the fibrin gels, HeLa cells were cultured in Transwell plates, in a confluent state. However, this provided very little surface for adherence, thereby hindering the favorable conditions for these epithelial cells, which are generally seeded in a monolayer and stop growing at confluency, due to contact inhibition. The amphiphilic aliphatic groups incorporated within the modified TBA were hypothesized to enhance the affinity between the fibrin scaffold and the cell membrane as a result of mimicry with the membrane lipids, therefore helping stabilization of the confluent cell culture. After culture for 48 h in the presence of fibrinogen, thrombin, and the active aptamer, the cells formed a closely packed layer, with no contact inhibition, since the scaffold provided a suitable surface for cell attachment. In contrast, cells cultured with inactive control oligonucleotides exhibited a non-uniform size circular shape and gaps in the cell layer. These results therefore confirmed the successful entrapment of modified TBA within the fibrin scaffold.

This methodology is a very interesting approach for in vitro tumor modeling and ex vivo tissue formation, amongst other applications. As described previously, 3D cultures for cancer modeling provide a much more realistic environment than monolayer cultures and a more ethical and less complex approach than in vivo assays [157]. Fewer than 5% of anticancer therapeutics complete clinical trials [158], probably because in vitro 2D cultures are not able to mimic the whole tumor architecture [159]. As such, this functionalized fibrin scaffold provides a suitable environment for epithelial–stromal interactions and mimics the tumor architecture, both of which are responsible for tumor progression, more accurately, thus providing a more accurate cancer model for in vitro tests.

SOEF can also enable entrapment and sustained release of therapeutic molecules. In a subsequent study, the same authors developed the SOEF technique further to efficiently deliver a camptothecin derivative (CPT1) to cancer cells [42]. Camptothecin is a natural apoptotic molecule that has been widely studied as an anticancer drug since it was first isolated in the early 1960s [160]. Furthermore, its derivatives, such as Topotecan or Irinotecan, have been approved by the FDA for a wide range of cancer treatments [161]. This approach involves a tandemly connected TBA moiety and a camptothecin-binding modified DNA aptamer (CMA), thus leading to a bifunctional aptamer (bApt) that allows CPT1 to be retained during gelation of the fibrin hydrogel. Fluorescent polarization assays showed that the amount of CPT1 trapped in the hydrogel was directly dependent on the initial fibrinogen concentration in the presence of equimolar concentrations of thrombin and CPT1/bApt complex. In vitro inhibitory assays were carried out in human tumor cell lines (Hela and HepG2 cells) and showed, in contrast to the free CPT1-containing hydrogels, a remarkable inhibition of cell proliferation in aptamer-containing hydrogels. A cytotoxicity assessment in non-cancerous cell lines could have been carried out as a complementary assay. The authors also compared the dependency of cell viability on CPT1 concentration with or without entrapment, and the resulting IC_50_ values showed a 170-fold lower concentration requirement for trapped CPT1, thus allowing a much lower dosage and, consequently, reducing side effects.

### 2.4. Silk

Silk is a unique protein-based polymer that is able to form fibers. Indeed, silk fibers are the strongest and toughest natural fibers known, which has encouraged research involving this biomaterial due to its multiple applications [162], for example in tissue engineering [163], drug delivery, or 3D bio-printing [164]. In nature, silk is synthesized by some Lepidoptera larvae, such as silkworms, spiders, scorpion, mites, and flies. Of these, cocoon silk from the domesticated silkworm (*Bombyx mori*) and the dragline and capture threads from large orb-weaving spiders (*Nephila clavipes*) are the most well characterized. Silkworms only produce one type of silk, which comprises two proteins, namely fibroin (structural) and sericin (from the family of glue-like proteins), during their lifecycle. In contrast, orb-weaving spiders are able to produce up to nine different silks, each of them using a different set of glands [165].

Silk is one of the most widely used natural biopolymers due to its high biocompatibility, biodegradability, and excellent mechanical properties [166]. Thus, a wide variety of silk-based biomaterials have been designed to develop multiple structures, such as hydrogels, sponges, or films, with potential applications in medicine. Furthermore, modification of the amino acid side chains allows tailor-made devices with tuned surface properties or immobilized growth or adhesion factors, such as aptamers for cell recognition, to be obtained [167].

#### Biosensing Application: Current Advances and Progress of Using Natural Protein-Based Polymers

The development of advanced depots for the storage and controlled release of active enzymes is one of the most innovative functions developed for silk–aptamer devices. This idea was explored by Humenik et al., who developed immobilized and functionalized DNA–spider silk nanohydrogels for selective inhibition of thrombin [36,168] (Figure 6). In this study, the authors developed nanofilms comprising recombinant spider silk proteins (rssp), which were functionalized with two different DNA aptamers, namely apt15, which inhibits thrombin proteolytic activity and has selective affinity for exosite I, and apt29, a heparin-binding regulatory site that can recognize thrombin exosite II [169]. Chemical attachment of both apt15 and apt29 to the rssp was carried out by taking advantage of bivalent ions, thus resulting in structural changes in the aptamer–rspp conjugates [170]. Circular dichroism (CD) spectroscopy and Size Exclusion Chromatography–Multiangle Light Scattering (SEC–MALS) analysis determined that the structures of these conjugates facilitated the targeting conjugate–thrombin bond and confirmed the selective aptamer-dependent inhibition of thrombin activity. AFM showed that a dense network comprising nanofibrils sown on nanofilms formed an immobilized nanohydrogel in water. Furthermore, FRET assays with co-assembled apt15–rssp and apt29–rssp hydrogels detected that, in the presence of thrombin-complementary oligonucleotides, the aptamer structure undergoes a conformational change and thrombin activity is reactivated.

In vitro clotting tests were performed using mixtures of thrombin and fibrinogen in a blood plasma model [171]. The analysis of turbidimetry results from these studies showed that aptamer–rssp nanohydrogels were able to inhibit thrombin activity in a specific way under physiological conditions. Thus, the use of aptamers with silk hydrogels represents an attractive approach for developing new biomedical devices that include sensitive biocatalysts.

Apart from silk-based hydrogels, more complex structures have been developed to take advantage of aptamer properties and silk in combination with other materials. Thus, Bendivi et al. recently published a new design for a complex aptasensor based on a glassy carbon electrode (GCE) modified with titanium oxide (TiO_2_) nanoparticles and a silk fibroin nanofiber (SF) composite [172]. Furthermore, the aptasensor included a DNA aptamer specifically designed to bind prostate-specific antigen (PSA). Although highly sensitive detection methods are used for early diagnosis of prostate cancer, including enzyme-linked immunosorbent assay (ELISA) and liquid chromatography–mass spectrometry (LC–MS), low-cost and highly sensitive protein measurements are still needed [173]. Among these new strategies, biosensors are a promising approach. Thus, Bendivi et al. designed and developed an electrochemical biosensor [174] for the early detection of prostate cancer. The incorporation of TiO_2_ nanoparticles and SF nanofibers allowed the surface area of the electrode to be increased. Moreover, SF improved aptamer binding by preventing nanoparticle aggregation, thereby achieving a homogeneous surface. The resulting aptasensor showed high selectivity, sensibility, and reproducibility, as demonstrated by the dynamic range and low detection limit for detection of PSA in serum samples. As such, this approach could form a solid basis for the development of electrochemical aptasensors with potential biomedical applications. However, despite its low cost and the small sample volume needed, further experimentation is still needed to reduce the time for both sensor preparation and PSA determination.

### 2.5. Hyaluronic Acid

Hyaluronic acid (HA), also known as hyaluronan, is a natural linear polysaccharide comprising the repeating disaccharide β-1,4-D-glucuronic acid-β-1,3-*N*-acetyl-D-glucosamine (Figure 7) [175,176]. HA is one of the main components of the extracellular matrix (ECM) and exhibits both mechanical and biological properties that mediate cellular signaling, matrix organization, embryonic development, angiogenesis, cell–matrix interactions during cell proliferation and migration, wound healing, and stem-cell differentiation, among others [176,177,178,179]. These important structural and functional roles in the body are promoted by the strong hydrophilic character and high molecular weight of HA [180]. Despite its ubiquitous presence, HA possesses outstanding relevance in the skin and is present in high concentration in synovial joint fluid, vitreous humor, hyaline cartilage, umbilical cord and intervertebral disc nucleus [181]. Due to its multiple advantages, such as its excellent biocompatibility, biodegradability, non-immunogenic role and ubiquitous expression in vertebrate tissues, HA forms the basis for many therapeutic scaffolds involved in bioengineering [182,183,184,185,186,187]. Furthermore, HA is widely used since its long polysaccharide chain possesses multiple functional groups, namely carboxylic acid, hydroxyl groups and the N-acetyl group, that allow covalent modifications and, therefore, the design of tailor-made biomedical products [182,188]. Consequently, HA is used for multiple biomedical applications, including regenerative medicine, tissue engineering, drug delivery, cell therapy, and three-dimensional (3D) cell culture [178,189,190,191,192,193,194].

#### 2.5.1. Glioma

One example of functionalized HA-based scaffolds for controlled delivery of therapeutic agents was designed by Wang et al., who developed a biofunctionalized HA microemulsion drug delivery system co-loaded with shikonin and docetaxel drugs to inhibit glioma growth [43]. Moreover, a single-stranded DNA aptamer (AS1411) was conjugated to specifically drive the dual drug co-loaded HA-based microemulsion to nucleolin, a protein overexpressed on the surface of glioma cell membranes [195,196]. As a result of the AS1411 aptamer, this drug delivery nanosystem decreased transepithelial electrical resistance, thereby increasing the apparent permeability coefficient in an artificial blood–brain barrier (BBB) model and suggesting that the targeted microemulsion was able to penetrate the BBB as a small nanoparticle (30 nm size). Furthermore, AS1411/SKN and DTX-M showed high cytotoxicity, enhanced cellular uptake, and triggered apoptosis-mediated cell death in U87 cells. In vivo assays demonstrated that AS1411/SKN and DTX-M selectively accumulated in the brain of tumor-bearing nude mice, retarded the growth of glioma and also prolonged the survival rate in an orthotopic glioma animal model. As such, these authors developed a co-delivery HA-based drug delivery system of shikonin and docetaxel, which could be a promising therapeutic approach for drug combinations as anti-glioma treatment.

#### 2.5.2. Axons Regeneration

Although typically used as selective targeting systems to specifically drive nano- or microdevices [197], aptamers can be also conjugated to natural biopolymers for subsequent release under controlled conditions. For example, Agrawal et al. designed and developed an oligonucleotide-functionalized HA hydrogel for release of the anti-nogo receptor (NgR) RNA aptamer in a controlled and sustained manner [44]. The anti-NgR aptamer had been previously demonstrated to compete with myelin inhibitors, namely Nogo-A, myelin-associated glycoprotein (MAG), and oligodendrocyte myelin glycoprotein (OMgp), for NgR binding [198,199], thereby triggering the regeneration of damaged axons after spinal cord injury [200]. The accuracy of this drug delivery system was tested by means of neurite outgrowth assays, and the results showed that increased binding affinity between the aptamer and the oligonucleotides enhanced sustained release of the aptamer for up to 28 days. Thus, the aptamer released successfully avoids the neuro-inhibitory effects of myelin-derived inhibitors and promotes in vitro neurite outgrowth from rat dorsal root ganglia. Consequently, as a result of the customized oligonucleotide design, the functionalized hydrogel could be applied to different injury and disease models.

### 2.6. Hyaluronic Acid Mixtures

Despite its multiple advantages, pure HA is difficult to apply clinically since it is easily degraded [201]. As such, new mixtures comprising HA and other polymers have been developed to expand the diversity of HA-based scaffolds. These hybrids are of considerable interest due to their potential to improve the mechanical properties of those comprising HA alone.

#### 2.6.1. Cartilage Repair

Hybrid hydrogels consisting of silk and HA have emerged as suitable scaffolds for biomedical applications in the last years [185,202,203]. Thus, Wang et al. developed a reinforced hydrogel using two different clinically available biomaterials, namely silk fibroin (SF) and HA, for cartilage tissue engineering in an osteochondral defect rabbit model [45]. Thus, the incorporation of HA could increase the chondrogenic ability of SF, a natural biopolymer from *Bombyx mori* widely used in cartilage restoration due to its excellent biocompatibility and mechanical strength [204], since HA has been shown to promote chondrogenesis both in vitro and in vivo [205,206]. Furthermore, the scaffold was functionalized with the Apt19s aptamer, which specifically recognizes pluripotent stem cells, to recruit endogenous mesenchymal stem cells (MSCs). The aptamer-functionalized scaffold was able to enhance cell adhesion and recruit bone-marrow-derived mesenchymal stem cells (BM-MSCs) in vitro. Implantation of the aptamer-functionalized scaffold into an osteochondral defect in vivo enhanced endogenous stem cell homing and promoted cartilage repair to a greater degree than the aptamer-free scaffold. As such, this functionalized scaffold approach could be a promising tissue engineering/regenerative medicine strategy for the treatment of chondral/osteochondral defects by means of aptamer-induced homing of MSCs.

Another hybrid HA-based scaffold comprising a poly(ethylene glycol) diacrylate/thiolated hyaluronic acid (PEGDA/tHA) mixture was studied to develop functionalized hydrogels containing an ssDNA aptamer against fibronectin [46], which is widely available in plasma and could mimic physiological conditions after implant insertion in a hypothetical surgery. Moreover, fibronectin is a promising coating for implantable biomaterials [207,208], since it provides a substrate for cell attachment in wound healing [209]. The presence of the aptamer enhanced the adhesion of human osteoblasts (hOBs) on hydrogels, as well as greater spreading, and more adhesion complexes were found compared to control hydrogels lacking the aptamer. Moreover, in vitro viability assays showed a significant increase in terms of number of cells present on the functionalized hydrogels. A high scaffold colonization was also achieved, since hOBs migrated deeper into the hyaluronan-based hydrogels in the presence of the anti-fibronectin aptamer and appeared on different focus planes. Thus, the authors demonstrated that enrichment of the biomaterial surface with the anti-fibronectin aptamer promoted cell adhesion and scaffold colonization.

#### 2.6.2. Cancer Therapy

Among the wide range of materials employed for development of drug nanocarriers, chitosan stands out due to its high water solubility, bioadhesive properties, and absorption-enhancing ability [210]. As such, a combination of HA and chitosan has been proposed to take advantage of the highly mucoadhesive properties of hyaluronan and the penetration-enhancing effect of chitosan, thereby achieving hyaluronan/chitosan nanoparticles (HACSNPs) [211]. Moreover, the inclusion of hyaluronan reduces non-specific interactions between HACSNPs and serum proteins [212]. For example, Ghasemi et al. developed aptamer-conjugated HACSNPs as nanocarriers for the targeted delivery of 5-fluorouracil (5FU) to mucin1 (MUC1)-overexpressing colorectal adenocarcinomas [47]. MUC1 is a large transmembrane glycoprotein the expression of which is at least 10-fold higher in malignant adenocarcinomas [213]. Physicochemical characterization showed a size of 181 nm and long-term stability, whereas in vitro assays in MUC1+ human adenocarcinoma (HT-29) and MUC1- Chinese hamster ovary (CHO) cells showed that both the cytotoxic effect and cellular uptake of nanoparticles were significantly higher than for the free drug, in both cell lines. Indeed, the effect was enhanced in the MUC1+ cell line. The sustained drug release and increased cytotoxicity of 5FU-loaded HACSNPs highlight the encouraging potential of HACSNPs as drug carriers for chemotherapeutic agents.

Another aptamer-targeted mixture involving chitosan and HA was developed by Varnamkhasti et al. (Figure 8) [48]. In this study, HA was used as the shell for chitosan nanoparticles to deliver conjugated SN-38, an active metabolite of the chemotherapeutic agent irinotecan, which is used clinically against colorectal cancer [214]. SN-38 has outstanding clinical relevance because this metabolite is up to 1000-fold more active than the original molecule [215], and conjugation to HA increases its stability and solubility. This HA-SN-38 conjugate was electrostatically linked to the positively charged chitosan nanoparticles, and the final nanosystem included surface decoration with a DNA aptamer specifically designed against MUC1. SEM micrographs showed spherical nanoparticles with a size of 125 nm and a rough surface, whereas in vitro release assays determined that SN-38 underwent enhanced cleavage in the acidic medium, which simulates the cancer cell microenvironment, as compared to physiological pH. These results could indicate that these HA-coated chitosan nanoparticles meet the requirements for controlled delivery in cancer cells, with reduced side effects in off-target tissues.

The presence of the MUC1 aptamer increased the uptake of nanoparticles by MUC1+ HT29 cells, as demonstrated by flow cytometry and confocal microscopy, whereas MUC1- CHO cells showed no enhanced uptake compared to non-targeted nanoparticles. Moreover, in vitro assays determined that the nanoparticles triggered apoptosis-mediated cytotoxicity. However, the adsorption of serum proteins affected the cytotoxic effect of targeted nanoparticles but did not significantly alter the cytotoxicity of the non-targeted nanoparticles. This fact could be explained by blockade of the aptamer accessibility and interference in the interaction between the aptamer and the target. Thus, further studies are needed to determine the effect of the protein corona on the targeting ability of HA-coated chitosan nanoparticles and its ability to reach targeted cells.

## 3. Conclusions

Despite the efforts that have been made, in different scientific fields, to improve the drug-delivery systems, regenerative medicine, and diagnostic technologies required for precision medicine, most current devices are based on single or very simple components. Personalized treatments require the combination of multiple building blocks that perform precise functions, such as recognition of a biomarker, responsiveness to the environment, delivery of therapeutics, and degradation at the desired rate.

Multifunctional modular materials can play a key role, as they can be designed and constructed to detect and dynamically communicate with the microenvironment, thereby achieving the required result. Adaptable biomaterials have been recently achieved by the addition of tunable components for sensing, as well as aptamers, due to their ability to bind various molecules with high affinity and specificity. Aptamer selectivity allows for the specific binding of different types of target, ranging from inorganic molecules to macromolecules or even entire cells. Naturally based biomaterials are currently being widely studied for their potential in the field of biomedicine (Figure 2), and it is expected that they will soon find widespread clinical use. Functionalization of these biomaterials with aptamers has been shown to result in improved candidates for precise medicine purposes.

Once functionality and safety can be guaranteed, the benefits from both a biomedical and potential market point of view, arising from the development of more efficient devices, will have a marked impact, as their improved efficiency, with respect to conventional materials, makes them some of the most promising products for biomedical systems.

The biomaterials described herein are obtained by following a biotechnological approach in which structural macromolecules are biofunctionalized by aptamer conjugation, thus providing interesting properties inherited from the natural molecules of origin. Components of the extracellular matrix, such as elastin, collagen, and hyaluronic acid, or structural proteins, such as silk, have been already considered in the past as excellent substitutes for biomedical purposes, due to their origin and properties, and further strategies to implement their properties, such as biofuntionalization or the selection of specific sequences via genetic engineering, have allowed for the development of various alternatives that can adapt to and meet the new challenges of precise biomedicine.

## Figures and Tables

**Figure 1 pharmaceutics-12-01115-f001:**
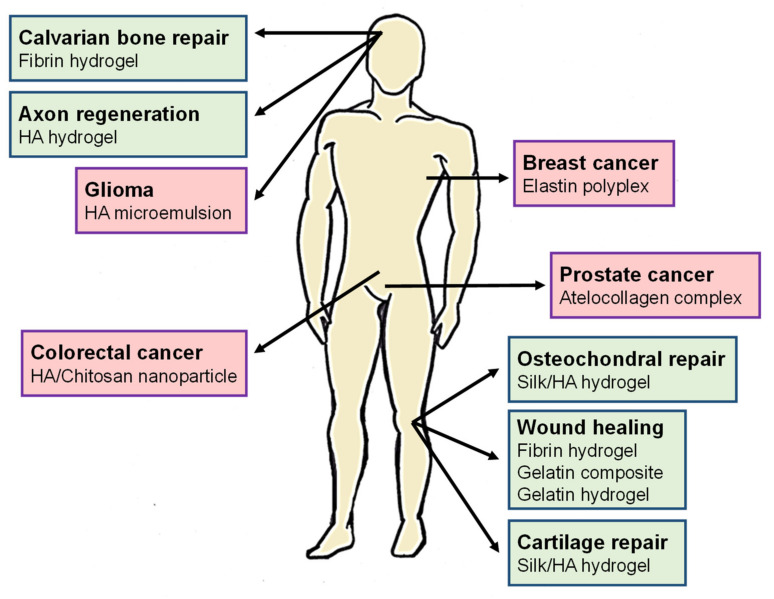
Scheme of the biomedical applications and their therapeutical approaches described in this review.

**Figure 2 pharmaceutics-12-01115-f002:**
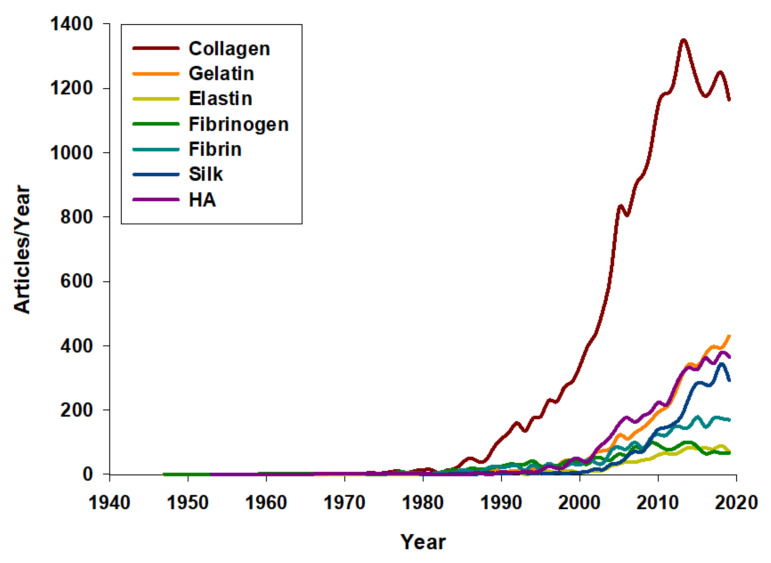
Scientific articles published referring to “X biomaterial” from 1940 to 2020. The number of articles published in each year, from 1940 to 2020, was identified by searching the terms referred in the legend in the PubMed database (https://pubmed.ncbi.nlm.nih.gov/), queried on 7 October 2020.

**Figure 3 pharmaceutics-12-01115-f003:**
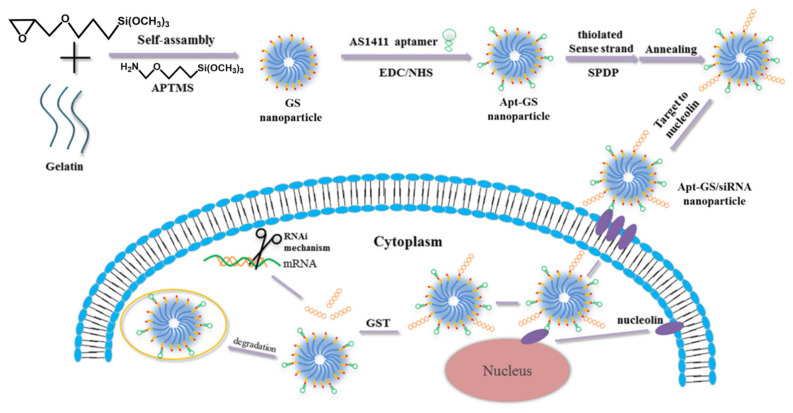
Scheme of the redox-responsive tumor-targeting Apt–gelatin–silica nanogels (GSNGs) developed for siRNA by Zhao et al. Reproduced from [22], Springer Nature, 2019.

**Figure 4 pharmaceutics-12-01115-f004:**
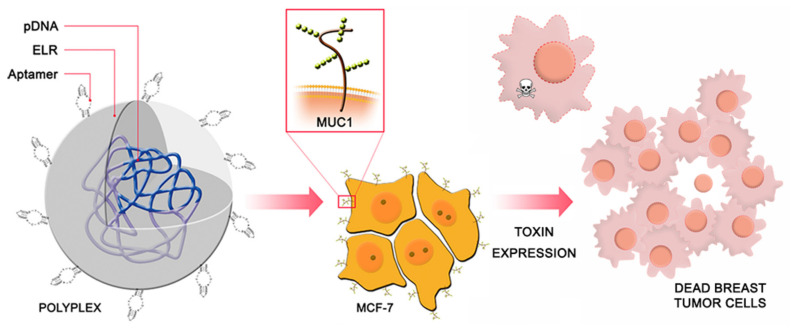
Scheme of the gene delivery system developed by Piña et al. Reproduced with permission from [30], American Chemical Society, 2016.

**Figure 5 pharmaceutics-12-01115-f005:**
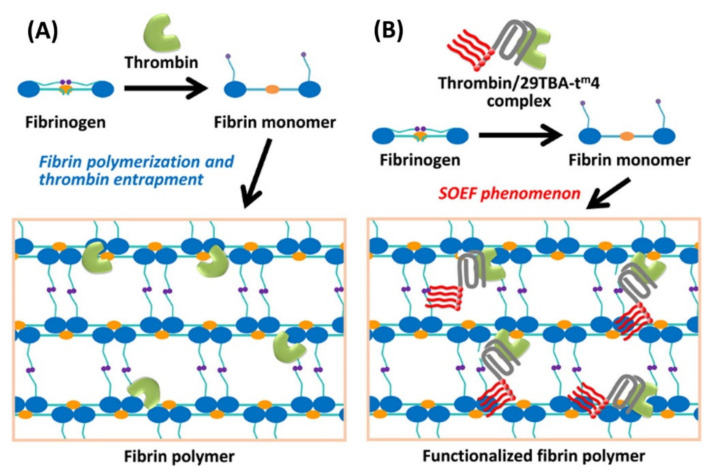
Scheme comparing fibrin formation and the “selective oligonucleotide entrapment in fibrin polymers” (SOEF) phenomenon for the preparation of a functionalized fibrin polymer using a modified thrombin-binding DNA aptamer. Reproduced with permission from [41], Elsevier, 2018.

**Figure 6 pharmaceutics-12-01115-f006:**
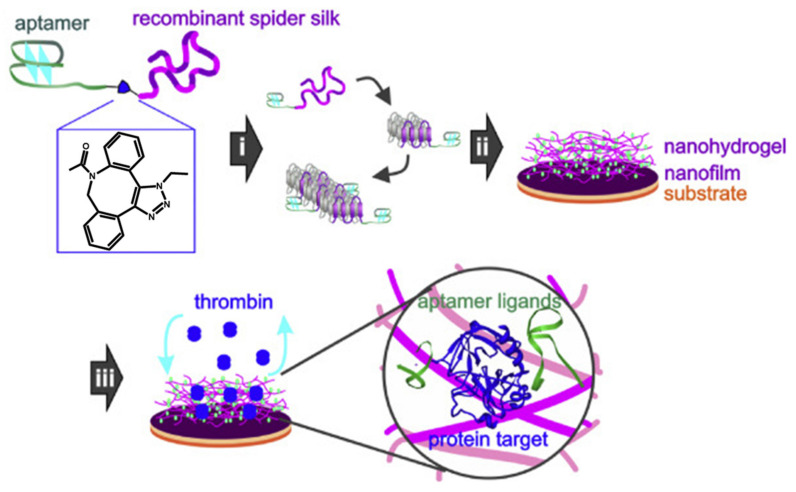
Scheme of the nanohydrogel assembly developed by Humenik et al., in Reference [36]. Adapted with permission from Elsevier, 2020.

**Figure 7 pharmaceutics-12-01115-f007:**
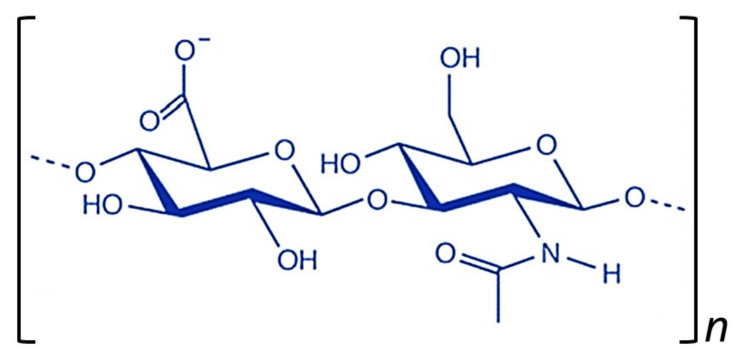
Molecular structure of the disaccharide unit of hyaluronic acid (D-glucuronic acid and *N*-acetyl-D-glucosamine).

**Figure 8 pharmaceutics-12-01115-f008:**
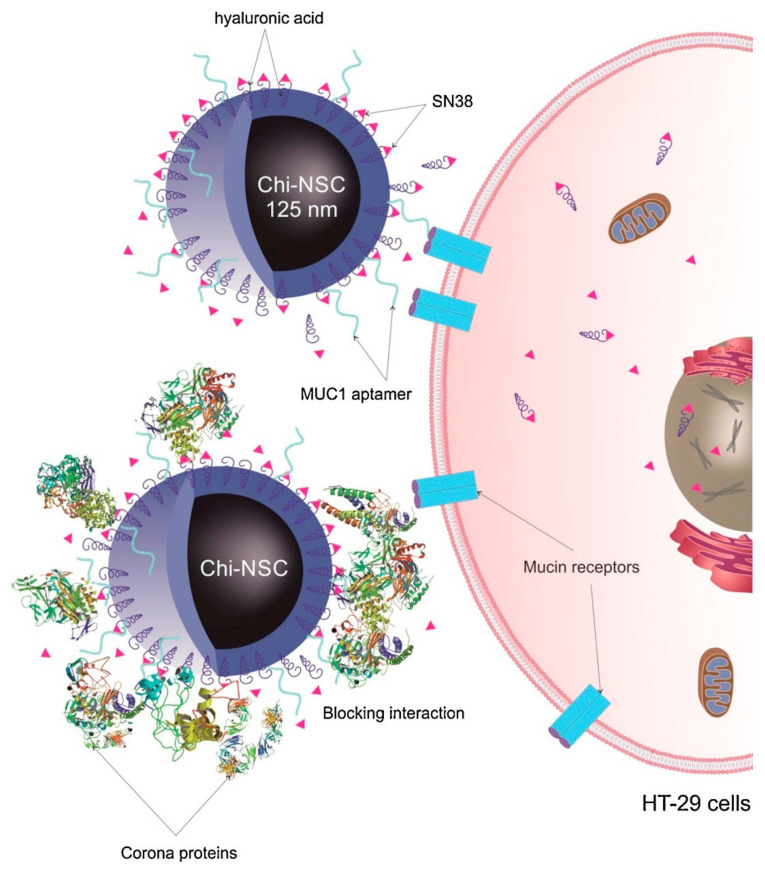
Scheme of interactions between bare and protein-coated MUC1 aptamer-conjugated HA/chitosan nanoparticles with MUC 1 positive HT-29 cell. Reproduced with permission from [48], Elsevier, 2015.

**Table 1 pharmaceutics-12-01115-t001:** A summary of different types of aptamer-targeted biomedical systems described in this review.

Biomaterial	Scaffold	Type of Aptamer	Target	Application	Reference
Atelocollagen	Complex	RNA	PSMA	Prostate cancer derived bone metastasis	[20]
Gelatin-Silica + PEG	Nanoparticles	DNA	Nucleolin	Gene delivery	[21]
Gelatin-Silica	Nanogels	DNA	Nucleolin	siRNA delivery	[22]
Gelatin	Composite	DNA	PDGF-BB	Molecule release	[23]
Gelatin-PEG	Hydrogel	RNA	VEGF	Cell and Growth factor sequestration	[24]
Gelatin	Nanoparticle assembly hydrogel	RNA	VEGF	Growth Factor sequestration and release	[25]
Collagen	Coating	DNA	Thrombin	Biosensing	[26,27]
Collagen-Graphene oxide	Composite	DNA	Dopamine	Biosensing	[28]
Gelatin	Coating	DNA	Chloramphenicol	Biosensing	[29]
Elastin	Polyplexes	DNA	MUC-1	Drug delivery	[30]
Elastin and PEG	Polyplexes	DNA	MUC-1	Drug delivery	[31]
Elastin and poly(aspartic acid)	Nanoparticles	DNA	MUC-1	Drug delivery	[32]
Elastin	Nanoparticles	DNA	MUC-1	Drug delivery	[33]
Elastin	Vesicles	RNA	DFHBI molecule	Visualization of cell-free gene expression	[34]
Elastin	Giant peptide vesicles	RNA	DFHBI molecule	Visualization of cell-free gene expression	[35]
Silk	Nanohydrogel	DNA	Thrombine exosite I and II	Selective and reversible inhibition of thrombin	[36]
Fibrin	Hydrogel	DNA	VEGF	Wound healing	[37]
Fibrin	Hydrogel	DNA	VEGF and PDGF-BB	Angiogenesis	[38]
Fibrin	Hydrogel	DNA	VEGF	Bone healing	[39]
Fibrin	Hydrogel		VEGF and PDGF-BB	MSC survival enhancement	[40]
Fibrin	Hydrogel	DNA	Thrombin	Molecule entrapment	[41]
Fibrin	Hydrogel	DNA	Thrombin + Camptothecin	Chemotherapy	[42]
HA	Microemulsion	DNA	Nucleolin	Glioma	[43]
	Hydrogel	RNA	NgR	Spinal cord injury	[44]
SF + HA	Hydrogel	DNA	MSCs	Cartilage repair	[45]
PEGDA + tHA	Hydrogel	DNA	Fibronectin	Tissue regeneration	[46]
HA + Chitosan	Nanoparticle	DNA	MUC1	Colorectal adenocarcinoma	[47]
	Nanoparticle	DNA	MUC1	Chemotherapy	[48]

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
