# Peer review of "Aptamer-Functionalized Natural Protein-Based Polymers as Innovative Biomaterials"

_pharmaceutics, 2020, doi:10.3390/pharmaceutics12111115_

Round 1

Reviewer 1 Report

This review paper discusses the application of aptamers in biomaterial. This is an interesting topic.

  1. However, I do find the paper is hard to read because of the structure of the review article. Since the review is on the aptamer applications in the biomaterials field, one would expect that the structure would be built around the applications, instead of the biomaterials. Because of the structure the authors used, I find the paper very hard to read, and often have to try to summarize for myself before moving on to the next biomaterial related topic. the structure issue must be addressed. 
  2. The authors talked about (bio)sensor. I do not think that they are examples of biomaterials, especially when the sensors are used for detections of drugs. biosensor applications are not considered biomaterials. 
  3. It is noted that strong statements are not supported by references. This has to be corrected. I am using one example: Page 2, 2nd paragraph: 

Different methodologies, ranging from extraction from natural sources to more modern advanced techniques, such as the recombinant production of synthetic
proteins that mimic their natural counterparts, can be used to produce natural-derived materials. The recombinant production of such materials, which include collagen, elastin, silk, gelatin, fibrinogen, and hyaluronic acid, amongst others, presents significant benefits in comparison with proteins extracted from animal tissue, especially their homogeneous composition, the absence of
animal infectious agents, or their lower immunogenic potential. The biodegradability of these materials allows their temporary use, when necessary, or their gradual replacement with newly formed tissue during regeneration. Moreover, the design and construction of synthetic genes offers
the possibility of combining unique functionalities found in nature in a modular, “Lego”-type strategy, thereby combining the complex characteristics of natural proteins with technological functionalities.

Author Response

Reviewer 1:

This review paper discusses the application of aptamers in biomaterial. This is an interesting topic.

1.However, I do find the paper is hard to read because of the structure of the review article. Since the review is on the aptamer applications in the biomaterials field, one would expect that the structure would be built around the applications, instead of the biomaterials. Because of the structure the authors used, I find the paper very hard to read, and often have to try to summarize for myself before moving on to the next biomaterial related topic. the structure issue must be addressed.

Answer: The authors are grateful for the reviewer’s comment. Although the manuscript is built around the functionalization of natural biomaterials with aptamers, subsections around the biomedical applications have now been included to make the review easy to read and to jump directly to the part of interest. Moreover, long paragraphs have been divided into smaller ones.

2.The authors talked about (bio)sensor. I do not think that they are examples of biomaterials, especially when the sensors are used for detections of drugs. biosensor applications are not considered biomaterials.

Answer: The authors also wanted to describe non-traditional fields of application of natural protein-based polymers conjugated with aptamers, being novel and less conventional approach for biomaterials of natural origin. In paragraph now called " Biosensing application: current advances and progress of using natural protein-based polymers", the authors have highlighted the possibility of their use also in alternative applications as well as a component or support in the development of new biosensors. The introduction of biological materials in certain biosensor devices, may be considered as an added value by increasing the biocompatibility and reducing possible adverse side effects.

3.It is noted that strong statements are not supported by references. This has to be corrected. I am using one example: Page 2, 2nd paragraph:

Different methodologies, ranging from extraction from natural sources to more modern advanced techniques, such as the recombinant production of synthetic proteins that mimic their natural counterparts, can be used to produce natural-derived materials. The recombinant production of such materials, which include collagen, elastin, silk, gelatin, fibrinogen, and hyaluronic acid, amongst others, presents significant benefits in comparison with proteins extracted from animal tissue, especially their homogeneous composition, the absence of animal infectious agents, or their lower immunogenic potential. The biodegradability of these materials allows their temporary use, when necessary, or their gradual replacement with newly formed tissue during regeneration. Moreover, the design and construction of synthetic genes offers the possibility of combining unique functionalities found in nature in a modular, “Lego”-type strategy, thereby combining the complex characteristics of natural proteins with technological functionalities.

Answer: The reviewer is right. 15 new references have now been added to support strong statements through the whole manuscript.

Reviewer 2 Report

The review is quite interesting as it reports on a hot topic. I suggest its publication.

I would just recommend to reorganize it a bit in order to make separate paragraphs. The MS contains very long paragraphs which would not allow the reader to jump directly to the part of his interest.

Author Response

Reviewer 2:

The review is quite interesting as it reports on a hot topic. I suggest its publication.

I would just recommend to reorganize it a bit in order to make separate paragraphs. The MS contains very long paragraphs which would not allow the reader to jump directly to the part of his interest.

Answer: The authors are thankful for the reviewer’ suggestion. The manuscript has been reorganized in new subsections related to the different biomedical application and diseases for each biomaterial type. Moreover, long paragraphs have been divided in smaller ones.

Reviewer 3 Report

In their review, Girotti et al. gave a rapid overview on aptamer-functionalized natural polymers prepared to be used as biomaterials.

The subject of this review is quite interesting. However, it would have been interesting to include more figures and/or schemes to illustrate the most important results which are described within this review. In addition, it seems strange that the Table 1 cited at the beginning of the review appears at the edn just before the conclusion. I suggest to the authors to move this Table 1 at the beginning of their review.

In view of this general comments and more specific ones given below, I do recommend the publication of this manuscript in Pharmacuetics after minor revisions. However, the authors have to answer to the questions raised by the reviewer and correct their manuscript as asked before publication.

Please find below specific comments which need to be addressed.

  1. Page 2 "The recombinant production ... with technological functionalities.": The authors have to give references for this part.
  2. Page 4, "mRNA": Even if it is obvious, the authors have to give the meaning of this abbreviation.
  3. Page 4 "Hao et al. have ... (PSMA)-targeting aptamer": How was PSMA conjugated to atelocollagen?
  4. Page 4 "another device ... for gene therapy [27].": What kind of cell-penetrating peptides are considered?
  5. Page 4 last line: "polyethylene glycol" has to be changed to "poly(ethylene glycol)".
  6. Page 5 "Additionally, ater ... Diferences between tissues.": And? To what was efficiency related to?
  7. Page 5 figure 2: the fromual are not visible. The authors have therefore to improve the quality of their figure.
  8. Page 6 "For exemple, ... free radical polymerization.": What are the monomers used for free radical polymerization?
  9. Page 6 "The nanoparticle suspension ... nanoparticle assembly.": I do not really understand what the authors want to explain.
  10. Page 6 "conjugation ... prior to lyophilisation.": How was conjugation performed?
  11. Page 6 "A several other studies ... to study cell survival.": I think that this sentence needs to be reworded.
  12. Page 7 "Given their ability ... dopamine or chloramphenicol.": The authors have to give references concenring this paragraph.
  13. Page 7 "To develop this biosensor ... by Au-S bonding.": What does "ssDNA" mean?
  14. Page 8 "In this scenario, ... proper biodistribution.": What are the nanoparticles considered in this study?
  15. Page 9. "The positive charge .. to be internalized by endocytosis.": The authors should give a range of what they consider as an appropriate particle size.
  16. Page 9 "Once a stable ... chemically conjugated.": How was the aptamer conjugated?
  17. Page 9 "The results demonstrated ... by the 5TR1 aptamer;": Is the increased transfection rate obserevd in all cell lines?
  18. Page 9 Figure 3: The quality of the polyplex drawing has to be improved.
  19. Page 10 "the first lock ... ripple effect in adjacent cells.": Is this paragraph linked to the previous one? Where is/are the corresponding reference(s)?
  20. Page 10 "The in vivo cancer ... cancer gene therapy.": Same question as the one just above. It seems that the answer is "yes", but it is not obvious.
  21. Page 10 "poly-aspartic acid": This word has to be changed to "poly(aspartic acid)" through all the manuscript.
  22. Page 10 "This problem was ... with the appropriate size.": What is the appropriate size?
  23. Page 10 "DLS showed the ability ... conjugated at their surface [98].": The term "appropriate size" doesn't mean anything. The authors have to specify what is the appropriate size for NPs.
  24. Page 11 "GFs are soluble proteins ... a role tissue regeneration [110].": I guess that a word is missing, probably "in".
  25. Page 11: What does "BB" mean?
  26. Page 12 "thiol-ene reaction": The authors have to precise between which compounds is realized.
  27. Page 12 "Upon addition of ... maintening its bioactivity.": Is the release of VEGF happen without a burst effect?
  28. Page 12 "Keratinocytes ... Ap-Fn hydrogels.": What are the controls used?
  29. Page 15 "CD spectroscopy and SEC-MALS ...": All the abbreviations used have to be defined.
  30. Page 15 Figure 5: The formula is not visible. The authors have to improve its quality.

Author Response

Reviewer 3:

In their review, Girotti et al. gave a rapid overview on aptamer-functionalized natural polymers prepared to be used as biomaterials.

The subject of this review is quite interesting. However, it would have been interesting to include more figures and/or schemes to illustrate the most important results which are described within this review. In addition, it seems strange that the Table 1 cited at the beginning of the review appears at the end just before the conclusion. I suggest to the authors to move this Table 1 at the beginning of their review.

Answer: The authors are grateful for the reviewer’s comments. 2 new figures have been included to illustrate a summary of the multiple applications and results described in this review and (Figure 1 and 8).

In view of this general comments and more specific ones given below, I do recommend the publication of this manuscript in Pharmaceutics after minor revisions. However, the authors have to answer to the questions raised by the reviewer and correct their manuscript as asked before publication.

Please find below specific comments which need to be addressed.

1.Page 2 "The recombinant production ... with technological functionalities.": The authors have to give references for this part.

Answer: The authors are thankful for the reviewer’s suggestion. 5 references have now been added to support strong statements through the whole manuscript.

2.Page 4, "mRNA": Even if it is obvious, the authors have to give the meaning of this abbreviation.

Answer: According to the reviewer’ suggestion the meaning of the abbreviation has now been included in the manuscript.

3.Page 4 "Hao et al. have ... (PSMA)-targeting aptamer": How was PSMA conjugated to atelocollagen?

Answer: The authors are thankful for the reviewer’s comment. In this work, Hao et al. performed chemical conjugation between maleimide-activated atellocollagen and SH-modified PSMA aptamer. This information is now included in the manuscript to facilitate the understanding of the work.

4.Page 4 "another device ... for gene therapy [27].": What kind of cell-penetrating peptides are considered?

Answer: The authors are thankful for the reviewer’s comment. The authors used TAT as cell penetrating peptide. This information is now included in the manuscript to facilitate the understanding of the work.

5.Page 4 last line: "polyethylene glycol" has to be changed to "poly(ethylene glycol)".

Answer: The reviewer is right. "polyethylene glycol" has been changed to "poly(ethylene glycol)" through all the manuscript.

6.Page 5 "Additionally, ater ... Diferences between tissues.": And? To what was efficiency related to?

Answer: The reviewer is right. The sentence was not clear and has been rewritten as follows: “However, in vivo luciferase expression quantification demonstrated that transfection efficiency did not correlate with in vivo biodistribution. Thus, while the highest accumulation was found in tumors and liver, gene expression was higher in heart. Furthermore, the GS-PEG/HA2-Apt nanogel could be leveraged for gene therapy in heart diseases, since gene expression was highest in heart tissue with lower accumulation”. These results could suggest that the modification of tumor-targeting moieties might be necessary to increase tumor gene expression specificity.

7.Page 5 figure 2: the formulas are not visible. The authors have therefore to improve the quality of their figure.

Answer: The reviewer is right. The authors have adapted the image and improved the quality of the formula.

8.Page 6 "For example, ... free radical polymerization.": What are the monomers used for free radical polymerization?

Answer: The authors are thankful for the reviewer’s comment. The monomers used for free radical polymerization are PEG, gelatin-MA and anti-VEGF aptamers, which has been added to the manuscript. This information is now included in the manuscript to facilitate the understanding of the work.

9.Page 6 "The nanoparticle suspension ... nanoparticle assembly.": I do not really understand what the authors want to explain.

Answer: The authors appreciate the reviewer’s suggestion. The nanoparticles suspended in solution undergo an assembly during the lyophilization process to form sponge-like structures. This explanation has been added to the manuscript.

10.Page 6 "conjugation ... prior to lyophilisation.": How was conjugation performed?

Answer: The authors are grateful for the reviewer’s comment. The conjugation process takes place between DBCO-modified GNPs and azide-modified anti-VEGF aptamer. This information is now included in the manuscript to facilitate the understanding of the work.

11.Page 6 "A several other studies ... to study cell survival.": I think that this sentence needs to be reworded.

Answer: The authors are thankful for the reviewer’s comment. The sentence has now been rephrased as follows: “According to previous studies demonstrating the ability of gelatin to stimulate cell adhesion [49, 50], the authors studied the synergistic effect of the gelatin scaffold and VEGF release.”.

12.Page 7 "Given their ability ... dopamine or chloramphenicol.": The authors have to give references concenring this paragraph.

Answer: The reviewer is right. The references for this paragraph have been added to the manuscript.

13.Page 7 "To develop this biosensor ... by Au-S bonding.": What does "ssDNA" mean?

Answer: The authors are thankful for the reviewer’ suggestion. The meaning of the abbreviation “ssDNA” for “single strand DNA” has now been included in the manuscript.

14.Page 8 "In this scenario, ... proper biodistribution.": What are the nanoparticles considered in this study?

Answer: The authors are thankful for the reviewer’ s comment. In this sentence, authors do not refer to a specific type of nanoparticles. The term “nanoparticles” has been changed by “nanodevices” to describe on the important advantages from a general point of view.

15.Page 9. "The positive charge .. to be internalized by endocytosis.": The authors should give a range of what they consider as an appropriate particle size.

Answer: The authors are grateful for the reviewer’ suggestion. The sentence has now been rewritten: “The positive charge of the ELR allowed complexion with the negatively charged plasmid DNA (pDNA) via electrostatic interactions to obtain a polyplex with a particle size of 140±20 nm, which is appropriate to be internalized by endocytosis (<200 nm) [107], [108].

16.Page 9 "Once a stable ... chemically conjugated.": How was the aptamer conjugated?

Answer: The authors are grateful for the reviewer’ s comment. The MUC1-targeted aptamer 5TR1 was covalently conjugated to the polyplex by electrostatic interactions between the positively charged polyplex and negatively charged DNA aptamer (5TR1).

17.Page 9 "The results demonstrated ... by the 5TR1 aptamer;": Is the increased transfection rate observed in all cell lines?

The results from Piña et al. demonstrated the excellent biocompatibility of polyplexes. An increase in the transfection rate was observed in the MUC-1 positive MCF7 cells, contrary to MUC-1 negative cells (MSCs, HUVEC, HFF1 and HepG2), due to the specificity provided by the 5TR1 aptamer.

18.Page 9 Figure 3: The quality of the polyplex drawing has to be improved.

Answer: The reviewer is right. The authors have included a higher quality image of the polyplex.

19.Page 10 "the first lock ... ripple effect in adjacent cells.": Is this paragraph linked to the previous one? Where is/are the corresponding reference(s)?

Answer: The reviewer is right. The corresponding reference has been added to the manuscript.

20.Page 10 "The in vivo cancer ... cancer gene therapy.": Same question as the one just above. It seems that the answer is "yes", but it is not obvious.

Answer: The reviewer is right. The corresponding reference has been added to the manuscript.

21.Page 10 "poly-aspartic acid": This word has to be changed to "poly(aspartic acid)" through all the manuscript.

Answer: The reviewer is right. "poly-aspartic acid " has been changed to " poly(aspartic acid) through all the manuscript.

22.Page 10 "This problem was ... with the appropriate size.": What is the appropriate size?

Answer: The authors are grateful for the reviewer’s comment. The sentence has now been rephrased as following: “This problem was resolved by mixing ELP-D-Gene A* and ELP-D with no Gene A* protein, which allowed the production of nanoparticles with a size of around 30 nm”.

23.Page 10 "DLS showed the ability ... conjugated at their surface [98].": The term "appropriate size" doesn't mean anything. The authors have to specify what is the appropriate size for NPs.

Answer: The authors are thankful for the reviewer’s comment. The sentence has been rewritten to add the size of the nanoparticles: “DLS showed the ability of the ELR-pRep system to form 40 nm nanoparticles with the DNA aptamer enzymatically conjugated at their surface [112]”.

24.Page 11 "GFs are soluble proteins ... a role tissue regeneration [110].": I guess that a word is missing, probably "in".

Answer: The reviewer is right. The missing word “in” has been added to the manuscript.

25.Page 11: What does "BB" mean?

Answer: The authors appreciate the reviewer’s comment. PDGF is a dimeric glycoprotein that can be composed of two A subunits (PDGF-AA), two B subunits (PDGF-BB) or one of each (PDGF-AB).

26.Page 12 "thiol-ene reaction": The authors have to precise between which compounds is realized.

Answer: The authors are thankful for the reviewer’s appreciation. The thiol-ene reaction takes place between acrylate-modified fibrinogen and SH-modified aptamer. This explanation has been added to the manuscript.

27.Page 12 "Upon addition of ... maintening its bioactivity.": Is the release of VEGF happen without a burst effect?

Answer: The authors appreciate the reviewer’s comment. The release happened in a sustained way due to the retention of VEGF by the aptamer.

In the native hydrogel 97% of VEGF was released in the first three days. On the contrary, hydrogels with aptamer released 35.7% of VEGF in the first hours, followed by a daily release rate of 3-4% between days 2 and 14. This information has been added to the manuscript.

28.Page 12 "Keratinocytes ... Ap-Fn hydrogels.": What are the controls used?

Answer: The authors are thankful for the reviewer’s comment. The controls used in the study are the bare fibrin hydrogel (Fn) and the fibrin hydrogel with scrambled aptamer and VEGF (Sc-Fn+VEGF)

29.Page 15 "CD spectroscopy and SEC-MALS ...": All the abbreviations used have to be defined.

Answer: The authors are thankful for the reviewer’ suggestion. The meaning of the abbreviation Circular dichroism (CD) spectroscopy and size exclusion chromatography-multiangle light scattering (SEC-MALS) have been added to the manuscript.

30.Page 15 Figure 5: The formula is not visible. The authors have to improve its quality.

Answer: The reviewer is right. The authors have adapted the image and improved the quality of the formula.

Reviewer 4 Report

The review about Aptamer-functionalized biomaterials is in my opinion interesting and well written.

References are updated

I think that the manuscript can be published 

Author Response

The authors thank the Reviewer

Round 2

Reviewer 1 Report

can be accepted.